# Discovering the neural correlate informed nosological relation among multiple neuropsychiatric disorders through dual utilisation of diagnostic information

## Abstract

The unravelled nosological relation among diverse types of neuropsychiatric disorders serves as an important precursor in advocating the dimensional approach to psychiatric classification. Leveraging high-dimensional abnormal resting-state functional connectivity, the crux of mining corresponded nosological relations is to derive a low-dimensional embedding space that preserves the diagnostic attributes of represented disorders. To accomplish this goal, we seek to exploit the available diagnostic information in learning the optimal embedding space by proposing a novel type of conditional variational auto-encoder that incorporates dual utilisation of diagnostic information. Encouraged by the achieved promising results in challenging the conventional approaches in low dimensional density estimation of synthetic functional connectivity features, we further implement our approach on two empirical neuropsychiatric neuroimaging datasets and discover a reliable nosological relation among autism spectrum disorder, major depressive disorder, and schizophrenia.

## 1 Introduction

### 1.1 Nosological relation among neuropsychiatric disorders

Instead of the traditional discrete, categorical view on nosology of multiple neuropsychological disorders (Frances, 2009), an alternative dimensional, continuous view, which suggests all mental illnesses lie along a single low dimensional spectrum (Adam, 2013; Casey et al., 2013; Helzer et al., 2009), uprises to be a more promising perspective to view the relation among diverse disorders. Despite its commonplace in the symptom-based clinical domain (Caspi & Moffitt, 2018), it was not until recently that the finding of shared genetic components across various neuropsychiatric disorders (Anttila et al., 2018) offered the first computational study to reveal the nosological relation among various disorders.

Aside from the discovered commonalities on the genetic makeup of various disorders, as the resting-state functional connectivity (FC) has proved itself as a valuable biomarker in distinguishing disorder patients from healthy controls in a plethora of computational studies (Woodward & Cascio, 2015), they may also hold great promise for uncovering the complex, continuous relation among different neuropsychiatric disorders. Unfortunately, due to their inherent high dimensionality, it poses a formidable data-analytic challenge in discovering the targeted nosological relation on the high-dimensional neuropsychiatric FC feature space. Thence, the crux of unveiling the FC informed nosological relation rests on finding an optimal low-dimensional embedding space to be informative of high-dimensional FC feature space.

### 1.2 The objective of this research

One straightforward approach is to apply dimensionality reduction techniques to these high-dimensional FC features. However, as stated in Huys et al. (2016), without the supplied supervision signals, over-reliance on data-driven methods may not be sufficient to derive the optimal

low-dimensional embeddings of FC features. Hence, the objective of this research is to seek the utilisation of supervision signals, e.g., diagnostic information, in learning a neural correlate informed low-dimensional embedding space, which allows us to attain the nosological relation among diverse disorders.

To harness diagnostic information in learning a targeted low-dimensional embedding space, in the following sections, we chiefly propose a novel type of conditional variational auto-encoder (VAE) to incorporate dual utilisation of this information, which involves regularising the learning of embeddings through the introduced implicit clustering effect, and encoding the diagnostic difference among disorders. Based on the demonstrated empirical superiority of our approach in a simulation study, we implemented it in mining a consistent nosological relation among three common disorders across two curated neuropsychiatric FC datasets.

In summarisation, main contributions of our work include:

- putting forward a novel type of conditional VAE, which involves two utilisations of diagnostic information in learning an optimal embedding space for high dimensional FC features;

- mining a consistent nosological relation among autism spectrum disorder, major depressive disorder, and schizophrenia across two curated neuropsychiatric FC datasets.

## 2 PREVIOUS WORKS

Harnessing the extracted brain features, recent efforts in computational psychiatry are largely acknowledged on developing increasingly complicated classification models to mine discriminative features to aid clinical diagnosis (Guo et al., 2017; Heinsfeld et al., 2018) and discovering the subtypes of a single mental illness via advanced data-driven approaches, e.g., the multi-view clustering method (Tokuda et al., 2018).

Unfortunately, little attention was paid to unveiling complex nosological relations among diverse types of neuropsychiatric disorders based on brain correlates. Among few attempts, a worth mentioned study that shares a similar motivation with ours is the work of Xia et al. (2018), where dimensions of psychopathology were closely tied to certain common brain features at the macro level, e.g., resting-state networks. However, this finding – in support of the differentiation of disorders on the brain feature level – reveals little on the nosological relationship among disorders, let alone their symptom-based methodology, which does not fully harness the essence of brain features.

Narrowing our focus on prevailing methodological modifications on VAE, a plethora of variants ranging from the improved generalisation, e.g., beta-VAE (Burgess et al., 2018), Vamp-VAE (Tomczak & Welling, 2018), the diversified generative process, e.g., EVAE (Bai et al., 2019), Multi-entity VAE (Nash et al., 2017), the optimised inference (encoding) process, e.g., Info-VAE (Zhao et al., 2017), the discretised latent space, e.g., vq-VAE (Oord et al., 2017), to the conditional variant and its recent advents, e.g., a semi-supervised VAE (Kingma et al., 2014), NVAE (Vahdat & Kautz, 2020), CVAE (Sohn et al., 2015), and the Gaussian mixture VAE (Dilokthanakul et al., 2016). Falling into the last category, our proposed method can be regarded as a novel type of conditional VAE that embeds a clustering concept. To interested readers, we attach a detailed methodological comparison between our approach and mentioned alternative approaches in A.1. Additionally, rather than the conventional usage in the work of Zhu et al. (2020), our implementation of the contrastive learning is directly applied in the embedding space instead of the original feature space.

## 3 DUAL UTILISATION OF DIAGNOSTIC INFORMATION

To exploit diagnostic information, i.e., the discrete diagnostic labels $c \in C$, in learning low dimensional latent variables in the embedding space $z \in \mathcal{Z}$ that are aware of both projected high dimensional FC input features $x \in \mathcal{X}$, and their diagnostic attributes, as shown in Figure 1, we delineate dual utilisation of diagnostic information in our proposed conditional VAE.

1. **The provided diagnostic information first serves as observable variables to regularise the learning of $z$ in a novel type of conditional VAE, ensuring the learned (low-dimensional) $z$ to be clustered around close-to-empirical cluster centers.**

2. **The $z$ associated diagnostic difference is further encoded in the embedding space via contrastive learning**.

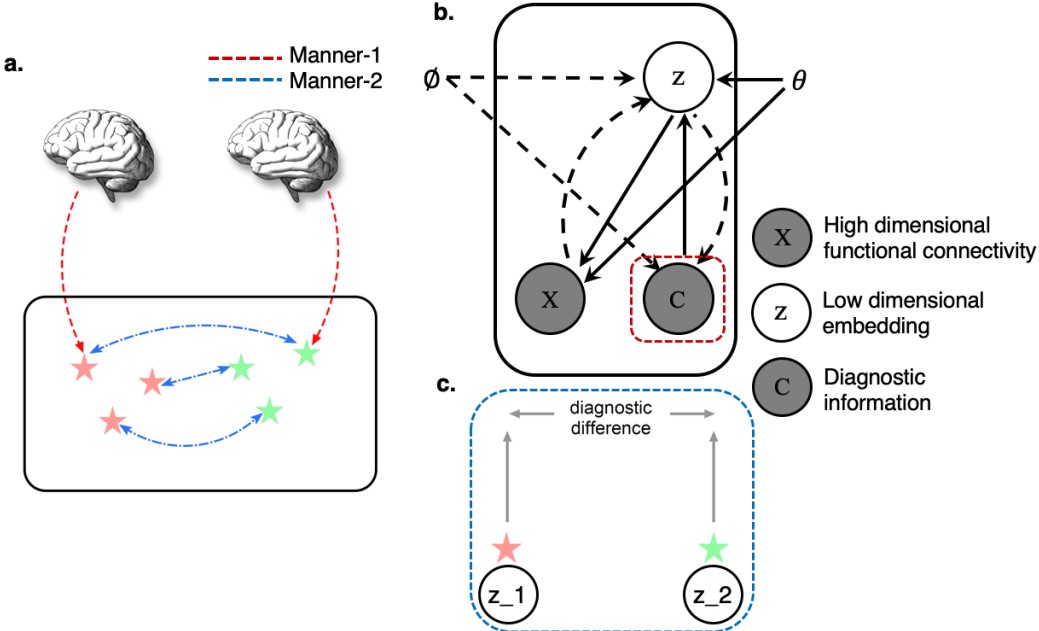

Figure 1: Dual utilisation of diagnostic information in a novel type of conditional VAE. **a.** The scheme of dual utilisation of diagnostic information. The red dotted line represents the Manner-1, i.e., provided diagnostic information serves as discrete variables to facilitate the projection of high-dimensional FC space into a low-dimensional embedding space (the red dotted line in subfigure **b**), whereas the blue dotted one denotes the Manner-2, i.e., the modelling of diagnostic difference in the projected low-dimensional embedding space. Both manners are incorporated into our derived novel type of conditional VAE. **b.** The graphical model rendering of our proposed novel type of conditional VAE, which encourages the encoded $z$ to be clustered in accord with their corresponded disorder phenotypes. Solid lines denote the generative processes; dashed lines refer to the inference processes. Unshaded circles denote the unobservable variables, whereas the shaded ones represent the observable variables. **c.** The thumbnail of the second manner: encoding the diagnostic difference in the embedding space through contrastive learning.

### 3.1 MANNER-1: REGULARISE THE LEARNING OF $z$ IN A NOVEL CONDITIONAL VAE

Under the coinage of VAE, an amortised joint inference distribution $q_\phi(z, x) \equiv q_\phi(z|x)p(x)$ is commonly employed to approximate the intractable joint generative distribution $p_\theta(z, x) \equiv p_\theta(x|z)p(z)$. Aside from the empirical distribution $p(x)$, we further assume $p(z)$ as a Normal distribution, and both $q_\phi(z|x)$ and $p_\theta(x|z)$ are conditional Gaussian distributions in which location and scale parameters can be parameterised by complex neural networks. The optimisation objective is acknowledged as the minimisation of the K-L divergence between two joint densities of latent variable and input, i.e., $p(z, x)$ and $q(z, x)$, as: $D_{KL}\{q_\phi(z, x)||p_\theta(z, x)\}$.

To harness provided diagnostic information, we explicitly add discrete variables $c \in C$ to represent available diagnostic information; it leads to a conditional VAE (Figure 1 (b)), which the general optimisation objective can be derived as $D_{KL}\{q_\phi(z, x, c)||p_\theta(z, x, c)\}$. Crucially, to further specify the inner components of two joint densities $q_\phi(z, x, c)$, and $p_\theta(z, x, c)$, we adopt a hierarchical encoding process that admits the following derivation: $q_{(\phi_z, \phi_c)}(c, z|x) \propto q_{\phi_c}(c|z)q_{\phi_z}(z|x)$. In a

similar vein, a hierarchical decoding process permits: $p_{(\theta_z,\theta_c)}(x,z|c) \propto p_{\theta_z}(x|z)p_{\theta_c}(z|c)$. As a result, the foregoing joint inference and generative distributions can be derived as: $q_{(\phi_z,\phi_c)}(x,z,c) \propto q_{\phi_c}(c|z)q_{\phi_z}(z|x)p(x)$ and $p_{(\theta_z,\theta_x)}(x,z,c) \propto p_{\theta_x}(x|z)p_{\theta_z}(z|c)p(c)$, respectively. From now on, we ignore subscripts for brevity. The general optimisation objective of this novel type of conditional VAE $\mathcal{L}_{cvae}$ with specified hierarchical inference and generative processes can be rewritten into a novel format:

$$
\begin{aligned}
&D_{KL}\{q_\phi(z,x,c)||p_\theta(z,x,c)\} \\
&= \sum_c \iint q(c|z)q(z|x)p(x) \log \frac{q(c|z)q(z|x)p(x)}{p(x|z)p(z|c)p(c)} dzdx \\
&= \mathbf{E}_{x\sim p(x)}\Big[\mathbf{E}_{z\sim q(z|x)}[-\log p(x|z)]\Big] + \mathbf{E}_{x\sim p(x)}\Big[\mathcal{D}_{KL}(q(c|z)||p(c)) + \sum_c q(c|z)\log\frac{q(z|x)}{p(z|c)}\Big],
\end{aligned}
\tag{1}
$$

where we ignore the optimisation irrelevant term $\log p(x)$ (an empirical distribution) for brevity. With $N$ number of classes, we define a categorical prior distribution $Cat(c|\pi), \pi \in \Re_+^N$ for $p(c)$. Three different parameterised Gaussian distributions are further defined as $p(x|z) = \mathcal{N}(x;\mu_x,I)$, $q(z|x) = \mathcal{N}(z;\mu_x,\sigma_x^2 I)$, and $p(z|c) = \mathcal{N}(z;\mu_c,I)$, respectively, where $I$ is the identity matrix. $q(c|z)$ can be modelled by a multi-layer feedforward soft-max network to produce a categorical representation (logits) for each encoded latent representation.

The first component in equation 1, i.e., $\mathbf{E}_{z\sim q(z|x)}[-\log p(x|z)]$, can be realised as the conventional reconstruction loss in a vanilla VAE to ensure the low-dimensional latent variables encoded from high-dimensional FC features. The entire second component, i.e., $\mathcal{D}_{KL}(q(c|z)||p(c)) + \sum_c q(c|z)\log\frac{q(z|x)}{p(z|c)}$, can be understood as the imposed (implicit) clustering effect in the low-dimensional embedding space.

The first term in the above-mentioned second component, i.e., $\mathcal{D}_{KL}(q(c|z)||p(c))$, serves as a preventative measure to prevent creating a dominant cluster in the face of a class-imbalanced dataset. Intuitively, it encourages the produced categorical representation (logits) $c$ from $q(c|z)$ to be close to its categorical prior distribution $p(c)$. However, since the K-L divergence is an intractable divergence measure, we compute an easy-to-measure alternative quantity, i.e., the cross-entropy of two distributions $H(q(c|z),p(c))$ in practice. Letting $N$ to indicate the number of input classes, the second term in the second component, i.e., $\sum_c q(c|z)\log\frac{q(z|x)}{p(z|c)}$, encourages the learned embedding space to be divided into $N$ different subspaces to ensure each encoded latent representation can be ascribed to 1 of $N$ clusters. Since the loss term that is on the clustering effect can be pinned down to $\frac{1}{2}||z - \mu_c||^2$ (see the derivation in attached A.2), where the cluster centre $\mu_c$ is provided directly from the precomputed empirical cluster mean, the first utilisation of diagnostic information can be regarded as enforcing each encoded embedding $z$ to be closer to its belonged cluster centre $\mu_c$ within each of $N$ clusters.

### 3.2 MANNER-2: ENCODING DIAGNOSTIC DIFFERENCE IN THE EMBEDDING SPACE

Corresponding to the second manner of utilising diagnostic information, i.e., embedding the diagnostic difference between disorder phenotypes, we resort to contrastive learning (Hadsell et al., 2006) to explicitly model this diagnostic difference. As shown in Figure 1(c), we define a pair of latent vectors $\overrightarrow{z_1}, \overrightarrow{z_2} \in Z$ to represent two encoded high dimensional brain features, a distance function $D$ to measure the dissimilarity of two latent vectors, and a binary class indicator $C = 0, 1$ to verify whether latent vectors come from the same disorder phenotype. The contrastive loss function $\mathcal{L}_{clr}$ (Hadsell et al., 2006) in our case can be defined as:

$$
(1-C)\frac{1}{2}(D_{(\overrightarrow{z_1},\overrightarrow{z_2})})^2 + C\frac{1}{2}\{\max(0, m - D_{(\overrightarrow{z_1},\overrightarrow{z_2})})\}^2,
\tag{2}
$$

where $m$ is a pre-definable margin to weigh the contribution of dissimilar pairs, and we adopt the simple euclidean distance to capture $D_{(\overrightarrow{z_1},\overrightarrow{z_2})}$, i.e., $D_{(\overrightarrow{z_1},\overrightarrow{z_2})} = ||\overrightarrow{z_1} - \overrightarrow{z_2}||_2$. To preserve the authentic diagnostic differences among encoded latent variables, the $m$ is set to 0, i.e., we weigh the equal importance of dissimilar and similar pairs.

Importantly, bridging the two loss terms, i.e., $\mathcal{L}_{clr}$ and $\mathcal{L}_{cvae}$ together, an overall loss function $\mathcal{L}_{total}$ through a convex combination of two terms can be derived as:

$$\mathcal{L}_{total} = (1 - \lambda)\mathcal{L}_{cvae} + \lambda\mathcal{L}_{clr}, \tag{3}$$

where we enforce the equal contribution of two loss terms, i.e., $\lambda = 0.5$, to stress the equal importance of two manners in utilising the diagnostic information to derive our targeted embedding space. In attached A.3, we show the effects of differentiated $\lambda$ in shaping the embedding space for interested readers.

## 4 ASSESSING THE IMPORTANCE OF DIAGNOSTIC INFORMATION

To assess the importance of diagnostic information in deriving the optimal low-dimensional embedding of neuropsychiatric FC features, we implemented our proposed method in a simulation study, along with the ground truth embeddings. The purpose of this simulation study is twofold. (1) We seek to assess the importance of diagnostic information in recovering ground truth embeddings. (2) We also aim to demonstrate the empirical superiority of our proposed approach over several existing supervised representation learning approaches.

In preparing for the synthetic high-dimensional FC features, we strictly follow the following three-step generation procedure.

1. We firstly pre-train a U-map projection function $\mathcal{F}$ on a real FC dataset, e.g., the SRPBS multi-disorder brain connectivity database (Tanaka et al., 2021).

2. 300 low dimensional embeddings are then sampled from three 2D multivariate Gaussians $\mathcal{N}_2(0.8, 1); \mathcal{N}_2(0.4, 1); \mathcal{N}_2(0.2, 1)$. The pair-wise distance relations among three 2D Gaussians are served as the ground-truth relation (see Figure 2).

3. The inverted projection function $\mathcal{F}^{-1}$ is further utilised to map the foregoing 300 embeddings from the low-dimensional (2D) ground truth embedding space into the high-dimensional FC space. In the case of the SRPBS dataset, this produces a $[300, 9730]$ synthetic FC feature matrix.

### 4.1 MODELS AND THEIR IMPLEMENTATIONS

#### 4.1.1 WITHIN OUR APPROACH

The crux of implementing our proposed approach is to compute the final loss function equation 3 with its involved loss terms $\mathcal{L}_{cvae}$ in equation 1, and $\mathcal{L}_{clr}$ in equation 2. Aside from the easy computed reconstruction loss, in the remaining components of equation 1, i.e., $\sum_c q(c|z) \log \frac{q(z|x)}{p(z|c)} + \mathcal{D}_{KL}(q(c|z)||p(c))$, the parameters that are pertaining to the optimisation of $\sum_c q(c|z) \log \frac{q(z|x)}{p(z|c)}$ can be reduced to $\sigma_x^2$, $\mu_x$, $z$, and $\mu_c$. As $\sigma_x$ and $\mu_x$ can be parameterised by a feed-forward neural network, it allows the production of $z$ via the re-parameteristion trick Kingma & Welling (2013), i.e., $z = \epsilon \bigotimes \sigma_x + \mu_x, \epsilon \sim \mathcal{N}(0, 1)$. The cluster-wise mean $\mu_c$ can be produced by a simple one layer Gaussian network with supplied supervision signals, e.g., cluster labels in this simulation study. A simple softmax network is sufficient to measure the KL-divergence between $q(c|z)$ and $p(c)$. The second loss term $\mathcal{L}_{clr}$ in equation 2 can be easily captured via adding a margin (distance) based feed-forward neural network on top of encoded latent representations.

Essentially, to demonstrate the importance of dual utilisation of diagnostic information in our proposed approach, corresponding to above-mentioned two utilisations, three instantiations of our approach are established ranging from the full implementation of our model (`Full model`) to the mere implementation of $\mathcal{L}_{cvae}$ (`Manner-1 model`), and $\mathcal{L}_{clr}$ (`Manner-2 model`). Their layer-wise configurations and respective parameter estimation processes are concisely summarised in attached B.1.

#### 4.1.2 ALTERNATIVE APPROACHES

Three categories of alternative methods are also included in this simulation study, ranging from the conventional semi-supervised VAE (`S-VAE`) (Kingma et al., 2014), the mainstream manifold learning based dimensionality reduction approach, e.g., supervised Umap (`S-Umap`) (McInnes et al.,

2018), to the metric learning based neighbourhood component analysis (NCA) (Goldberger et al., 2004). The implementation details of these alternative approaches are also demonstrated in attached B.1.

## 4.2 Performance evaluation

As shown in Figure 2, provided with ground truth embeddings and their pairwise distance relations, we target on evaluating the performance of various models regarding the capability of recovering the ground truth embeddings from synthetic high-dimensional FC features. The evaluation metrics include the qualitative visual inspection of yielded embedding spaces, and a quantitative assessment to verdict whether the distance relations among estimated clusters are preserved.

To quantify the between-cluster distance on the derived low dimensional embedding space, we rely on the pair-wise Wasserstein distance-based Frechet inception distance (FID) based metric (Heusel et al., 2017), i.e., $FID_{(\mu_1, \Sigma_1, \mu_1, \Sigma_2)}$, over the conventional $L-1$ and $L-2$ distance metrics for its consideration on both first and second degree statistics in measuring the discrepancy between two multivariate Gaussians. Its computation can be defined as:

$$FID_{(\mu_1, \Sigma_1, \mu_1, \Sigma_2)} = |\mu_1 - \mu_2|^2 + tr(\Sigma_1 + \Sigma_2 - 2(\Sigma_1 \Sigma_2)^{1/2}),$$

where $\mu; \Sigma$ are the first and second degree statistics of inferred low-dimensional multivariate Gaussians, i.e., $z \sim \mathcal{N}(\mu, \Sigma)$. On the basis of suggested FID metric to assess the between-cluster relation, the provided ground-truth relation among three clusters can be expressed into the three-way inequality: $FID_{(Cluster1-Cluster3)} > FID_{(Cluster1-Cluster2)} < FID_{(Cluster2-Cluster3)}$.

Juxtaposed in demonstrated Figure 2, among three instantiations of our approach, the maximum utilisation of diagnostic information, i.e., the `Full model`, achieved the most promising result on recovering the correct distance relation among three instantiations of our approach, along with the assessed significant pair-wist t-tests among three between-cluster distances in our `Full model` (see attached C).

This observation, in conjunction with the consistent result in the second simulation study (see attached D), greatly substantiates the importance of diagnostic information in deriving an optimal low-dimensional space for high-dimensional FC like features. The superiority of our approach over included alternative ones are presented less-distorted low-dimensional embedding space and recovered accurate distance matrix. This observed empirical superiority of our approach assures the further implementation on real neuropsychiatric FC datasets.

## 5 Mining the nosological relation among multiple disorders

After demonstrating the importance of diagnostic information in learning low-dimensional embeddings of FC features, we seek to implement our approach, i.e., the `Full model`, on curated neuropsychiatric FC datasets to explore nosological relations among disorder phenotypes. In short, high-dimensional FC features of included neuropsychiatric disorders are firstly projected into a common low-dimensional (2D) embedding space, where their pairwise distances are served as the nosological relation in this embedding space.

## 5.1 Datasets & Metrics

To acquire a high-quality neuropsychiatric FC dataset that contains multiple types of disorders is a time-consuming task, which demands expertise in diagnosis, and consistency in neuroimaging scanning protocols. The SRPBS multi-disorder brain connectivity database (Tanaka et al., 2021) [1] that pools the fMRI scans of diverse neuropsychiatric patients from 8 different scanning sites with consistent diagnosis guidelines, perfectly serves as our research database.

**UTO dataset**    From the opted SRPBS database, two datasets were formed to suit our need in studying the nosological relation among multiple neuropsychiatric disorders. The primary dataset comes from the data that were acquired at the University of Tokyo Hospital (AKA., UTO dataset), which is

---

[1]The URL link for this database can be referred to `https://bicr-resource.atr.jp/srpbsfc/`.

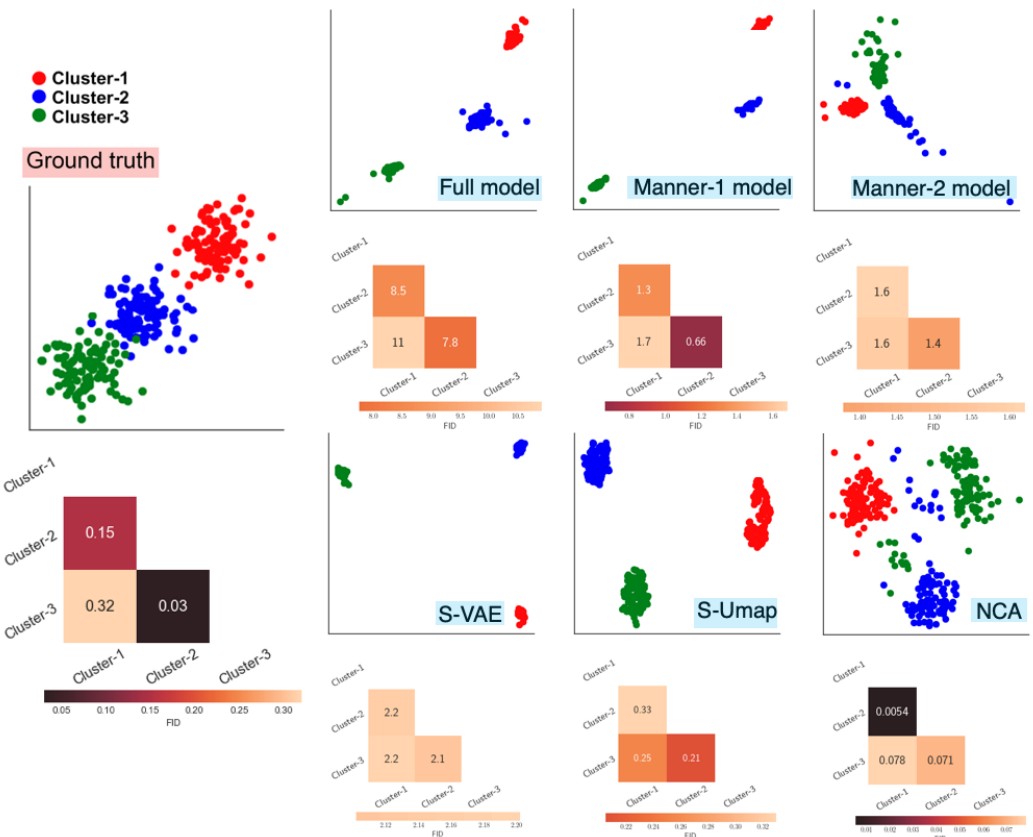

Figure 2: Model performance on synthetic FC features. The attached distance matrix is based on the computed FID distance between clusters in the derived low-dimensional embedding space. For each evaluated model, both the learned embedding space and the computed distance matrix were averaged over 20 runs of implementation.

comprised of fMRI scans from 3 chosen types of clearly diagnosed disorders: autism spectrum disorder (ASD; 10 subjects), major depressive disorder (MDD; 62 subjects), and schizophrenia (SCZ; 35 subjects). These fMRI scans were acquired in the same facility under the identical scanning protocol, which is free from the common threat of site difference in most fMRI studies. The brief introduction of the SRPBS database can be found in attached E.

The subject-wise brain features, i.e., functional connectivities, were obtained from 10-min resting-state fMRI BOLD signals that underwent the identical preprocessing steps. To compute the subject-wise functional connectivities from these preprocessed BOLD signals, the Brainvisa Sulci Atlas parcellation scheme, which each individual image was divided into 140 regions (Perrot et al., 2011) and the standard Pearson R correlation method were adopted for time series extraction, and the computation of connectivity matrix [2], respectively. Hence, for each subject, 9730 connectivity features were crafted to serve as high dimensional brain features, forming a $[107, 9730]$ brain feature matrix.

**HuShoWa dataset** Aside from the main UTO dataset, a replication dataset from the SRPBS database were further curated from the Hiroshima University and ShoWa University (HuShoWa dataset). A total of 307 subjects were included in our HuShoWa dataset, occupying categories of MDD (173 subjects), ASD (115 subjects), and SCZ (19 subjects). Underwent the identical preprocessing steps, and the feature extraction process, we harvested a $[307, 9730]$ FC feature matrix. To minimise the effect of site-difference on acquired fMRI scans in HuShoWa dataset, we resorted to the harmonisation technique, e.g., the combat approach (Yu et al., 2018) to correct site-difference in our input features.

---

[2]To ease the computation, the lower diagonal of connectivity matrix is reserved for connectivity features.

**Validation scheme & performance assessment**    Since empirical neuropsychiatric FC features are seldom accompanied with ground truth low-dimensional embeddings, it is essential to obtain a strict validation scheme to ensure the reliability of our attained nosological relations. For this, we opt for 10-fold cross validation scheme that demands 10 times running of our model on two empirical datasets.

In consist with the previous harnessed FID metric in assessing the between-cluster relation, in both UTO and HuShoWa datasets, we applied the FID metric to compute the cluster-wise FID distance on the basis of yielded low-dimensional embeddings of three disorders, e.g., $FID_{(ASD-MDD)}; FID_{(ASD-SCZ)}; FID_{(MDD-SCZ)}$, their inequality may inform us the nosological relation among ASD, MDD, and SCZ. Over the implemented 10-fold cross validation, multiple paired t-tests in testing the statistical significance of these pair-wise FID distances were also conducted to assess the reliability of attained nosological relations.

## 5.2    Discovered nosological relationships among ASD, MDD and SCZ

As shown in Figure 3 (left panel), on low-dimensional embedding spaces that are derived from the UTO and HuShoWa datasets, with maximum utilisation of diagnostic information, the learned embeddings are compact, clear-cut clustered in accordance with their represented disorder phenotypes [3].

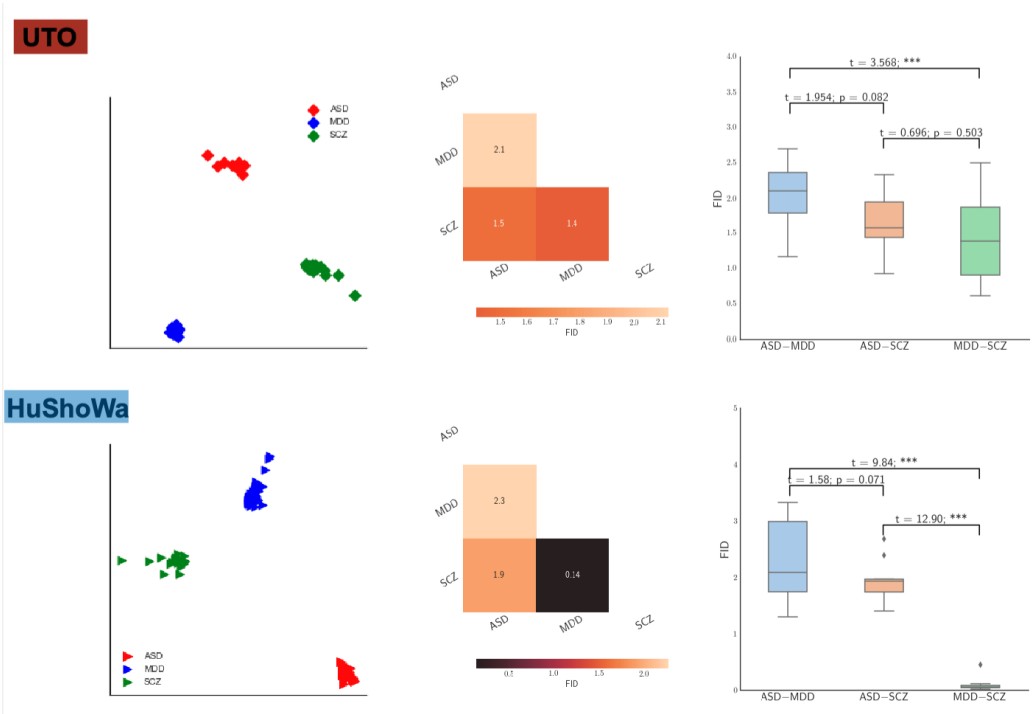

Figure 3: The discovered nosological relation among ASD, MDD, and SCZ across the UTO and HuShoWa datasets. Left panel: learned embedding spaces that are derived from the test data of UTO and HuShoWa datasets through the average over 10-fold cross validation. Middle panel: the computed FID distance matrices based on the averaged FID distances over 10-fold cross validation. Right panel: the conducted paired t-tests on pairs of FID relation of three disorders in 10-fold cross validation. The training configuration for this empirical experiment is recorded in attached B.2.

In respect of the attained highly consistent embedding spaces (Figure 3 left panel), two similar FID matrices (Figure 3 middle panel), and high statistical power in conducted paired t-tests (Figure 3 right panel) across two curated neuropsychiatric datasets, it unveils us an interestingly between-phenotype relation among the considered ASD, MDD and SCZ, i.e., $FID_{(ASD-MDD)} >$

---

[3]To interested readers, in attached F, we demonstrate the empirical comparison of the derived embeddings from our approach and several competing alternatives on HuShoWa dataset.

$FID_{(ASD-SCZ)}$ and $FID_{(ASD-MDD)} > FID_{(MDD-SCZ)}$. This suggests that ASD and MDD may belong to two independent nosology entities, which are loosely connected regarding their large differences in corresponding neural correlates (functional connectivities), whereas the closely linked ASD-SCZ, MDD-SCZ nexuses indicate the potential 'nosological closeness' between SCZ and the other two disorder phenotypes, which is in line with the previous findings on high rates of comorbidity on SCZ-ASD (Wood, 2017) and SCZ-MDD (Tsai & Rosenheck, 2013).

Based on this finding, we are able to align these disorder phenotypes on an FC informed dimensional coordinate, allowing ASD and MDD to situate at two ends of this coordinate, in correspondence with two domains of psychopathology, i.e., the neurodevelopmental and affective pathology, respectively (see attached G) (Craddock & Owen, 2010).

### 5.3 NEUROSCIENTIFIC INSIGHT

Furthermore, since the final projected FCs were represented as two-dimensional features, it is of increasing research interest to see whether these two-dimensional features are in agreement with existing disorder related connectivities in the brain space. For this, harnessing the layer-wise relevance propagation method (Montavon et al., 2019), we have projected each 'cluster centre' (the most representative feature in each of three clusters) back into the high dimensional FC space, and tentatively identified a significant connection between `Caudate` and `Cuneus` that obtains the highest relevance in discriminating three disorders (MDD, SCZ, ASD) among the overall 9730 FCs. However, the validity of this preliminary neuroscientific insight on involved functional connectivity is under the threat of used nonlinear projections (multiple ReLU connections in both inference and generative networks), where the isomorphism between the low-dimensional embedding and high-dimensional FC space is not guaranteed.

Lastly, to assess the interpretability of learned low-dimensional embedding space in our context, we focus on probing the potential association between learned embeddings and measured clinical or psychological assessments. On the basis of provided ASD related autism spectrum quotient (AQ score) (Baron-Cohen et al., 2001), and MDD related Beck Depression Inventory (BDI score) (Beck et al., 1996) in the HuShoWa dataset, the associations between our derived embeddings and the clinical factors are captured, i.e., AQ score: $\rho = 0.533, p < 0.001$ and BDI score: $\rho = 0.540, p < 0.001$, suggesting the potential clinical awareness of derived low-dimensional embeddings.

## 6 LIMITATIONS

Two limitations of this study are the confined capability of our approach in discerning a novel subtype of a disorder or the nosological overlap between two disorders, and the scarcity of high-quality independent FC datasets that are curated other than the harnessed SRPBS database. The former limitation may be ameliorated through investigating the finer (possibly hierarchical) structure within low-dimensional embeddings of known phenotypes, whereas the latter one may be tackled by the future release of the high-quality imaging datasets from the Brain/Minds beyond MRI project (Koike et al., 2021).

## 7 CONCLUSION

Unveiling the complex nosological relation among diverse neuropsychiatric disorders with the help of diagnostic information, we advocate a neural correlate based approach, which targets on deriving a neural correlate informed low-dimensional embedding space. To learn low-dimensional embeddings of neural correlates that preserves the diagnostic attributes, dual utilisation of diagnostic information that is incorporated in a novel type of conditional VAE, are proposed in this research. Relying on this method, we discovered a reliable and consistent nosological relation among ASD, MDD, and SCZ across two curated neuropsychiatric FC datasets.

Undoubtedly, the current approach – at its embryonic stage -- faces several mentioned limitations, a future approach with enhanced interpretability on the obtained low dimensional embedding space, and the inclusion of an independent validation dataset should enable us to attain more reliable nosological relations on various neuropsychiatric disorders to encourage the age-old, discrete categorical view on mental illness to make an exit at a fast pace.

## ETHICS STATEMENT

Concerning the potential ethnic issues of our propose approach in serving as a clinical application, we raise four concerns on future use of our approach.

1. Undoubtably, the sole reliance on the discovered nosological relation on basis of our approach is neither optimal nor responsible in the diagnosis procedure. Given the early stage of our approach, harnessing the discovered nosological relation among disorders is still immature to replace established symptom-based diagnosis in clinical psychiatry.

2. Regarding discovered nosological relations among disorders, given the heterogeneous, complex, and developmental nature of most neuropsychiatric disorders, further validation of such relations is demanded.

3. Meanwhile, since our probed nosological relation among disorders on 2D embedding space is induced from the computed distance relation, it does not imply any potential signs of comorbidity.

4. To yield a wide range of social acceptance on using neuroimaging modality in clinical diagnosis, it demands more future validation studies that come from other aspects, e.g., the genetic aspect, and symptom relevance. Solely relying on the neuroimaging modality in clinical diagnosis can be misleading and unethical.

Unless these raised ethical concerns are addressed, along with additional validation studies and evidence, clinical use of our approach should not be considered and recommended at this moment.

## REPRODUCIBILITY STATEMENT

To boost the reproducibility of our study, the executable Python code snippet that allows the implementation of our model is contained within the uploaded Zip file. Within the uploaded Zip file, we also include a code snippet that contains the helper function we used in the computation of FID scores between embeddings. The used SRPBS database can be referred to `https://bicr-resource.atr.jp/srpbsfc/`. The public available version of this referred database can be requested upon the site.

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

# A    METHODOLOGICAL DISTINCTIVENESS & DERIVATIONS

## A.1    METHODOLOGICAL COMPARISON BETWEEN OUR APPROACH AND ALTERNATIVES

As summarised in following Table 1, the main difference between our approach and NVAE (Vahdat & Kautz, 2020) lies on the derived hierarchical process in defining the decoding and generative process. In NVAE, the hierarchical process can be seen in the sequential production of $z$, i.e., $z = z_1, ..., z_l; q(z_2|x) = q(z_2|z_1, x)$. This sequential encoding and decoding procedure allows NVAE to implement feature combinations from $z_1$ to $z_l$. Differ to NVAE, our hierarchical process focuses on the conditional generation of the latent variable $z$ from both discrete variable $c$, and the input $x$, i.e., $q(c, z|x) \propto q(z|x)q(c|z)$. More importantly, with a defined categorical prior distribution for the discrete variable $Cat(c|\pi), \pi \in \Re_+^N$ for $p(c)$, the $N$ groups of $z$ are encouraged to be mutually independent, i.e., the implicit clustering effect. Such independence is neither the objective nor the outcome of NVAE approach.

The hierarchical process in our inference model entails the conditional chain $q(c|z)q(z|x)p(x)$, whereas in CVAE (Sohn et al., 2015), such conditional chain is replaced by the conditional density $q(z|x, y)$. In a similar vein, as illustrated in Table 1, the major distinction of our approach to a conventional semi-supervised VAE (Kingma et al., 2014) also lies on their differentiated derivations on inference networks.

In regards to different instantiations on graphical models, our approach also differs from the Gaussian mixture VAE on both inference and generative networks. In Gaussian mixture VAE (Dilokthanakul et al., 2016), the inference network is defined as $q(x, w, z|c)$, and its corresponded generation network is $p(c, x, w, z) \propto p(x|w, z)p(c|x)$. Ignoring the Gaussian mixture variable $w$, the foregoing networks can be seen as $q(x, z|c) \propto q(z|x)q(x|c)$ and $p(x|z)p(c|x)$. In our networks, these two processes are defined as $q(c|z)q(z|x)$ and $p(x|z)p(z|c)$, respectively. Aside from these differences, we acknowledge the explicit modelling of different Gaussian mixtures $w$, which cannot be achieved by the current version of our approach.

TABLE 1: MODELLING DIFFERENCES BETWEEN OUR APPROACH AND ALTERNATIVES.

| MODEL | GENERATIVE NETWORK | INFERENCE NETWORK |
|---|---|---|
| NVAE | $\Pi_l p(z_l|z_{<l})p(x|z_l)$ | $\Pi_l q(z_l|z_{<l}, x)p(x)$ |
| CVAE | $p(z|x)p(c|x, z)$ | $q(z|x, c)p(x, c)$ |
| SEMI SUPERVISED-VAE | $p(x|z, c)p(z|c)p(c)$ | $q(z|x, c)p(x)p(c)$ |
| GAUSSIAN MIXTURE VAE | $p(x|z)p(c|x)p(z)$ | $q(z|x)q(c|z)p(x)$ |
| OUR APPROACH | $p(x|z)p(z|c)p(c)$ | $q(c|z)q(z|x)p(x)$ |

## A.2    DERIVATIONS

In Section 3.1, we put forward a novel type of conditional VAE that allows implicit clustered embeddings. The core component of our loss function is pertaining to $\log \frac{q(z|x)}{p(z|c)}$, where we embed the implicit clustering effect. Given two conditional Gaussians to parameterise $q(z|x)$ and $p(z|c)$, respectively, i.e., $q(z|x) \sim \mathcal{N}(\mu_x, \sigma_x^2)$, and $p(z|c) \sim \mathcal{N}(\mu_c, I)$, the general optimisation loss term

for this component can be derived into:

$$\log \frac{q(z|x)}{p(z|c)} = \log \frac{\mathcal{N}(\mu_x, \sigma_x^2)}{\mathcal{N}(\mu_c, 1)}$$

where $\mathcal{N}(\mu_x, \sigma_x^2) = \frac{1}{\sqrt{2\pi\sigma_x^2}} \exp\left\{-\frac{1}{2}\left\|\frac{z - \mu_x}{\sigma_x}\right\|^2\right\}$, and $\mathcal{N}(\mu_c, 1) = \frac{1}{(2\pi)} \exp\left\{-\frac{1}{2}\left\|z - \mu_c\right\|^2\right\}$,

$$= \frac{1}{2}\|z - \mu_c\|^2 - \frac{1}{2}\log\sigma_x^2 - \frac{1}{2}\left\|\frac{z - \mu(x)}{\sigma(x)}\right\|^2, \text{ as } z = \mu_x + \epsilon \otimes \sigma_x, \epsilon \sim \mathcal{N}(0, 1)$$

$$= \frac{1}{2}\|z - \mu_c\|^2 - \frac{1}{2}\log\sigma_x^2 + \frac{1}{2}\|\epsilon\|^2.$$

Ignoring the last constant term $\frac{1}{2}\|\epsilon\|^2$, we arrive at

$$= \frac{1}{2}\|z - \mu_c\|^2 - \frac{1}{2}\log\sigma_x^2.$$

(4)

From the following general derivation, it clearly indicates the contribution of $\mu_c$ in producing our targeted low dimensional embeddings $z$. I.e., from the regularisation viewpoint, $\mu_c$ can be viewed as one type of regulariser to regularise the learning of $z$.

### A.3 THE IMPORTANCE OF $\lambda$ IN DETERMINING THE SHAPE OF $z$

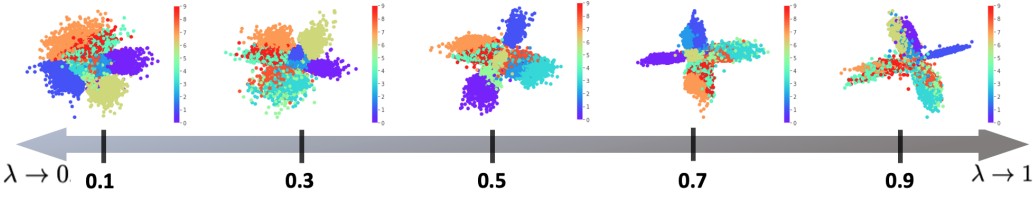

Figure 4: The role of $\lambda$ in our proposed approach. With the increment on the value of $\lambda$, the cluster-wise distances are enlarged, along with the increasing amount of void space among clusters.

In the main text, to balance the contribution of two manners, we pre-define the $\lambda = 0.5$ to enforce the equal weighting between two manners. However, this $\lambda$ value be tuned flexibly in determining the shape of each embedded cluster. In the attached Figure 4, through the demonstration on MNIST dataset, we show the effect of differentiated $\lambda$ in controlling the overlapping degree of encoded clusters. I.e., $\lambda \to 0$, clusters of embeddings tend to be overlapped with each other, whereas $\lambda \to 1$, encoded clusters are pulling away from each other, in the contribution of the independent, non-overlapping clustering effect.

## B TRAINING CONFIGURATIONS

### B.1 CONFIGURATIONS IN THE SIMULATION STUDY (SECTION 4.)

The following two subsections are presented here to demonstrate the default configurations for implementing three instantiations of our proposed approach, i.e., `Full model`, `Manner-1 model`, and `Manner-2 model`, and three alternative models, i.e., `S-VAE model`, `S-Umap model`, and `NCA model`, which were utilised in Section 3 of the main text.

Table 2: Model configurations of three instantiations of our proposed approach.

| Instantiation | Configurations |
|---|---|
| Full model | Network-1 ($z$) #1: FC 9730;200;2;200,9730, ReLU activation
Network-2 ($c$): FC 4000; 2000;100, ReLU and Soft-max activations
Network-3 ($\mathcal{L}_{clr}$):FC 40, ReLU activation, $D$: Euclidean distance
$\lambda = 0.5$, Batch Size: 20, num of batch: 20
optimiser:Adam, (Kingma & Ba, 2014) |
| Manner-1 model | Network-1 ($z$):FC 9730;200;2;200,9730, ReLU activation
Network-2 ($c$): FC 4000;2000;100, ReLU activation
Batch Size: 20, num of batch: 20
optimiser:Adam (Kingma & Ba, 2014) |
| Manner-2 model | FC 40, ReLU activation
$D$: Euclidean distance, Batch Size: 20, num of batch: 20
optimiser:Adam (Kingma & Ba, 2014) |

Table 3: Model configurations of three alternative approaches

| Model | Configurations |
|---|---|
| S-VAE model | Network-1 ($z$): FC 500, ReLU activation
Network-2 ($c$): FC 500, FC 9730, ReLU and Sigmoid activations
Network-3 ($\mu_c$): FC40;3, ReLU and softmax activation
Batch size: 20, num of training epoch: 20
optimiser: rmsprop, (Hinton et al., 2012) |
| S-Umap | num of components: 3, num of neighbours: 10
space: Euclidean space
num of training iterations: 1k, convergence tolerance: $1e^{-4}$ |
| NCA | num of components: 3,
initialisation approach: LinearDiscriminantAnalysis (Duda et al., 2006)
num of training iterations: 1k, convergence tolerance: $1e^{-4}$ |

## B.2 CONFIGURATIONS IN THE EMPIRICAL STUDY (SECTION 5.)

The following table summarises the training configuration of our approach in exploring the nosological relations among various disorder phenotypes in two empirical datasets (Section 4 of the main text). To improve the reliability of the attained nosological relation, we adopt the nearly identical model configurations to implement our model in UTO and HuShoWa datasets, respectively. Regarding the scarce sample sizes of two datasets, the complex cross validation procedure may not be appropriate here. Therefore, in Figure 4 of the main text, the results were yielded from two full datasets.

Table 4: Model configurations in UTO and HuShoWa datasets.

| Dataset | Configurations |
|---|---|
| UTO
& HuShoWa | Network-1 ($z$) #1: FC 9730;400;200;100;2;100;200,1000;9730, ReLU activation
Network-2 ($c$): FC 4000; 2000;100, ReLU and Soft-max activations
Network-3 ($\mathcal{L}_{clr}$):FC 40, 20, ReLU activation, $D$: Euclidean distance
$\lambda = 0.5$, Batch Size: 1024, num of batch: 500
optimiser:Adam, (Kingma & Ba, 2014)
number of training epochs: 80
batch size: 256
the usage of early-stopping: No |

## C    STATISTICAL TESTS

To assess the statistical significance of attained embedding relation in the above-mentioned simulation study (see Main text Section 4.), we conduct a series of paired t-tests to assess the significance of pair-wise between-cluster FID across six different approaches. As shown in the following Figure 5, our proposed `Full model` achieved the most superior t-test results among the harnessed 6 approaches, albeit the comparison between C2-C3 and C1-C2 does not achieve the statistical significance. Note here, the observed strong statistical significance between C2-C3 and C1-C2 in NCA, i.e., $C2 - C3 > C1 - C2$, contradicts with the ground-truth, i.e., $C2 - C3 < C1 - C2$.

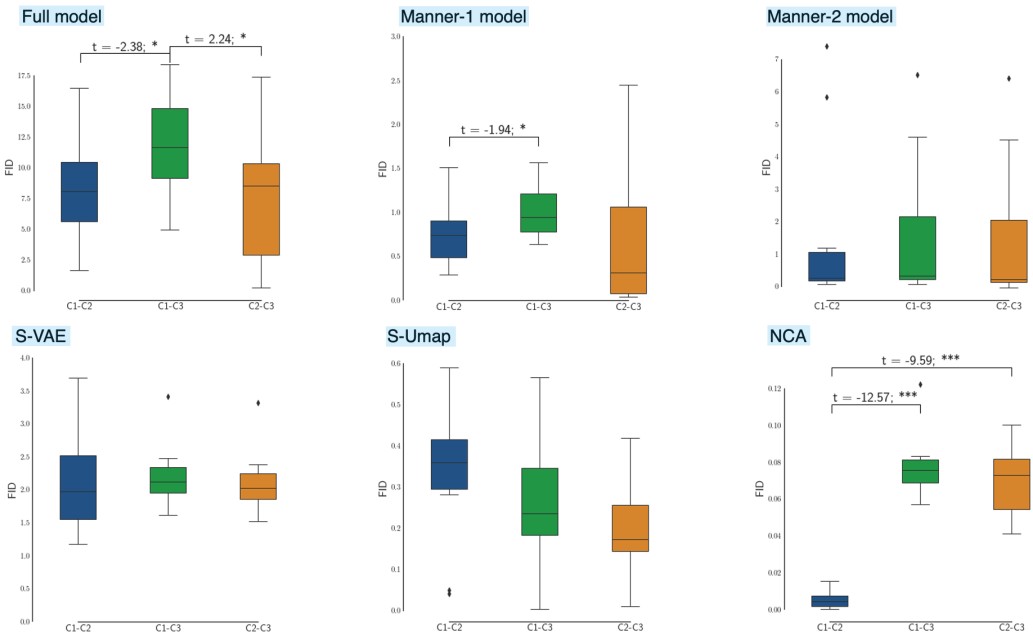

Figure 5: Paired t-tests in assessing the statistical significance among the computed between-cluster distances across 20 runs of implementation. All reported significant t-test scores are Bonferroni corrected. Only t-test scores that are statistical significant, are reported in this figure. Abbreviation index: C1/C2/C3: Cluster1/2/3.

## D    THE SECOND SIMULATION STUDY

We attach a supplementary (2nd) simulation study (see the following Figure 6) to further solidify the observed empirical superiority of our approach, i.e., the full model, in recovering low-dimensional embedding of FC-like high-dimensional features. In this attached 2nd simulation study, all set-ups are kept the same with the 1st simulation study (Section 4. main text) except for the non-linear scattering of simulated ground-truth embeddings. Inconsistent with the rendered 1st simulation study, the only model that recovers the ground-truth between-cluster relation is our proposed `Full model`.

## E    SRPBS DATABASE & DATA PRE-PROCESSING STEPS

The employed UTO and HuShoWa datasets are derived from the SRPBS database, which is consisted of resting-state fMRI scans from 8 different sites. The detailed curation and pre-precessing steps of SRPBS database can be consulted in the work of Tanaka et al. (2021). Here, we highlight some of the key aspects of scanning and data pre-processing. At each site, the subject-wise resting-state fMRI data were obtained from 10-min resting-state fMRI scanning session with the eye-open condition. For data curated in HuShoWa dataset, the imaging data were acquired in a Siemens Spec-

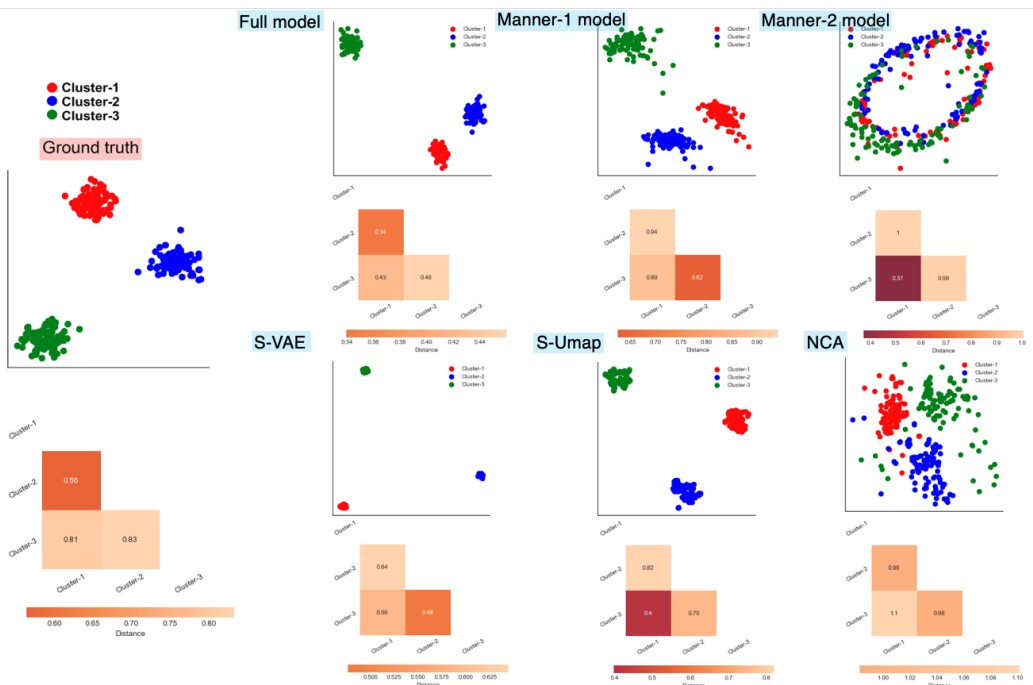

Figure 6: The second simulation study on synthetic FC features. In line with the 1st simulation study in the main text, we compute the distance matrix on the basis of between-cluster FID in the derived low-dimensional embedding space. The training configuration of each model is identical to the 1st simulation study.

tra and a Siemens Verio scanner, whereas for the data in UTO dataset, data were acquired from a GE MR750W scanner.

All resting-state fMRI data underwent the identical pre-processing step include slice-time correlation, realignment, co-registration, segmentation of T1-weighted structural images, normalisation to Montreal Neurological Institute (MNI) space, and spatial smoothing with an isotropic Gaussian kernel of 6 mm full-width at half-maximum.

# F  EMPIRICAL COMPARISON OF DERIVED EMBEDDINGS FROM OUR APPROACH AND ALTERNATIVE MODELS

Implemented on the same HuShoWa dataset, we empirically evaluate the derived low dimensional embeddings from our approach, and several alternative ones, e.g., neighbourhood component analysis (NCA) (Goldberger et al., 2004), T-SNE (Van der Maaten & Hinton, 2008), Supervised Umap (S-Umap) (McInnes et al., 2018), and a conventional semi-supervised VAE (S-VAE) (Kingma et al., 2014).

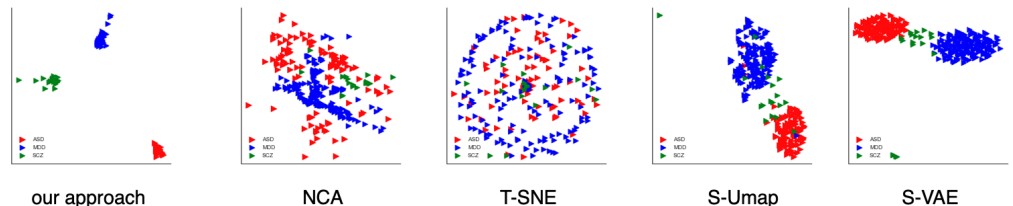

Figure 7: Empirical comparison of embeddings derived from our approach and several alternative approaches on HuShoWa dataset.

## G   DIMENSIONAL VIEW ON ASD, MDD, & SCZ

Demonstrated in the following Figure 8, in accordance with the envisioned dimensional view on diverse neuropsychiatric disorders (Craddock & Owen, 2010), we recast the attained the nosological relation among ASD, MDD, SCZ on the following continua, where ASD and MDD reside at two poles of this continua.

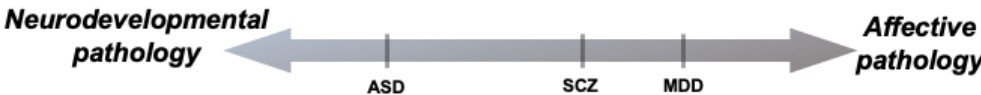

Figure 8: The dimensional view on ASD, MDD and SCZ

