# OpenReview forum: "Discovering the neural correlate informed nosological relation among multiple neuropsychiatric disorders through dual utilisation of diagnostic information"
_ICLR.cc/2022/Conference — ICLR 2022 Submitted_

### Official Review · Reviewer_nZ9a · 2021-10-31

**Correctness:** 3
**Technical Novelty And Significance:** 3
**Empirical Novelty And Significance:** 2
**Recommendation:** 8
**Confidence:** 4

**Main Review:**

First of all I would like to thank the authors for this interesting piece of work. Below, I will briefly list the main strength and weaknesses of this submission and make suggestions for improvement:

Strengths
-	The authors provide a useful extension to existing work on VAEs, which appears to be well-suited for the target application they have in mind.
-	The authors include both synthetic and empirical data as test cases for their method and compare it to a range of related approaches.
-	I especially appreciated, that the authors validated their method on the empirical data and also provide an assessment of face validity using established psychological questionnaires (BDI and AQ).
-	I also appreciated the ethics statement pointing out that the method requires additional validation, before it may enter the clinic.
-	The paper is to a great extend clearly written.

Weaknesses
-	In Figure 2 it seems that Manner-1 use of diagnostic information is more important than Manner-2 use of this information, which calls your choice into question to set lambda = 0.5 in equation 3. Are you able to learn this parameter from the data?
-	Also in Figure 2, when applying your full model to the synthetic data, it appears to me that inverting your model seems to underestimate the within-cluster variance (compared to the ground truth). Could it be that your manner-1 use of information introduces constraints that are too strong, as they do not allow for this variance?
-	It would strengthen your claims of “superiority” of your approach over others, if you could provide a statistical test that shows that your approach is indeed better at recovering the true relationship compared to others. Please, provide such tests.
-	There are important information about the empirical study missing that should be mentioned in the supplement, such as recording parameters for the MRI, preprocessing steps, was the resting-state recorded under eyes-open or eyes-closed condition? A brief explanation of the harmonization technique would also be appreciated. It would also be helpful to mention the number of regions in the parcellation in the main text.
-	The validation scheme using the second study is not clear to me. Were the models trained on dataset A and then directly applied to dataset B or did you simply repeat the training in dataset B. If the latter is the case, I would refer to this as a replication dataset and not a validation dataset (which would require applying the same model on a new dataset, without retraining).
-	Have you applied multiple testing correction for the FID comparisons across diagnoses. If so which? If not, you should apply it and please, state that clearly in the main manuscript.
-	It is somewhat surprising that the distance between SCZ and MDD is shorter than between SCZ and ASD as often the latter two are viewed as closely related. It might be helpful to discuss, why that may be the case in more detail.
-	The third ethics statement is not clear to me. Could you clarify?
-	The font size in the figures is too small. Please, increase it to improve readability.


**Summary Of The Paper:**

The authors present a new conditional VAE, which uses diagnostic information as a supervision signal in two different ways, with the aim to identify small latent embedding spaces for functional brain connectivity that preserve diagnostic information. The authors have potential applications in computational psychiatry in mind, which are indeed extensively discussed in the field (transitioning to a continuous nosological approach to mental illness). They compare their approach on synthetic data to various related approaches and ultimately apply it to two functional connectivity datasets from patients with different psychiatric diagnoses, which appears to give good results.

Their main contribution is technical and lies in defining a new cost function to learn the embedding space. Their work incrementally builds on existing work, but appears to be sufficiently novel. There is also a moderate empirical contribution as they analyze the relationship between MDD, ASD and SCZ.

**Summary Of The Review:**

Overall, I believe this to be a solid paper. There are some remaining issues that need addressing, before I can raise my score, for example pertaining to reporting and ensuring that rigorous statistics and quantifications are used to substantiate the authors' claims. If these comments are adequately addressed, I am sure, that this work would be a valuable contribution that hopefully will also be of interest to other research areas.

---

### Official Review · Reviewer_Cbpb · 2021-11-02

**Correctness:** 2
**Technical Novelty And Significance:** 1
**Empirical Novelty And Significance:** 1
**Recommendation:** 1
**Confidence:** 5

**Main Review:**

1. The English of this paper is poor which makes it difficult to read.
2. I do not understand the purpose of this paper, for example, what is the use of the low-dimensional representation of the functional connectivity obtained by the algorithm? Is it to classify patients? However, using label information to guide dimensionality reduction obviously introduces data leakage problems.
3. The author tries to use diagnostic information to guide the dimensionality reduction of functional connections. However, unlike other diseases, one of the main problems of mental illness is that there is no gold standard for the diagnosis, which relies on the description of subjective symptoms. This means that diagnostic information cannot guide the analysis of brain imaging data.


**Summary Of The Paper:**

This paper presents a method to reduce the dimensions of resting-state functional connectivity for psychiatry disorders such as autism spectrum disorder, major depressive disorder, and schizophrenia. Their method is based on a conditional variational auto-encoder that utilized the diagnostic label. They evaluated their method on two neuroimaging datasets and obtained clustered 2-dimensional representations of multi psychiatry disorder.

**Summary Of The Review:**

Extensive evidence showed that these neuropsychiatric disorders share symptoms and neuroimaging features. For example, several cross-disorder studies showed that major depressive disorder (MDD), bipolar disorder (BD) and schizophrenia (SZ) share biological, neuropsychological and clinical features, despite the criteria for their respective diagnoses being different. (Chang, Miao, et al. "Identifying and validating subtypes within major psychiatric disorders based on frontal–posterior functional imbalance via deep learning." Molecular psychiatry 26.7 (2021): 2991-3002)( Zhang, Luheng, et al. "Three major psychiatric disorders share specific dynamic alterations of intrinsic brain activity." Schizophrenia research (2021).) Therefore, the utilization of diagnostic information will not help to separate different disorders. Furthermore, these mental illnesses are often misdiagnosed. For example, the diagnosis of 50.7% of mental disorder participants was changed in a 10-year follow-up study, including schizophrenia spectrum disorders, bipolar disorder, major depression disorder. (Bromet, Evelyn J., et al. "Diagnostic shifts during the decade following first admission for psychosis." American journal of psychiatry 168.11 (2011): 1186-1194.)
Therefore, I believe the clinical motivation of this paper is wrong.

---

### Official Review · Reviewer_94nu · 2021-11-02

**Correctness:** 3
**Technical Novelty And Significance:** 2
**Empirical Novelty And Significance:** 2
**Recommendation:** 5
**Confidence:** 3

**Main Review:**

STRENGTHS:

1. The paper is well written, self-contained, and easy to follow. The methodology and main technical contributions are clearly articulated and explained. Moreover, the authors demonstrate experimental results over two separate clinical datasets independently.

2. As per my knowledge, the clinical problem of interest, i.e. a dimensional mapping of various psychiatric disorders (as opposed to diagnostic classification) is a relatively less studied problem in (data-driven) functional connectivity analysis. The methodology of the paper could serve as a tool to expand the psychiatric characterization offered by functional connectivity analysis for clinical applications.

WEAKNESSES:

1. Clinical Motivation:

The authors motivate the need for their framework by comparing their problem of interest with the categorical characterization (case/control) of neuropsychiatric disorders. Given the lack of such a  “ground truth” dimensional characterization available apriori, the results are hard to draw strong conclusions from. Moreover, motivation wise, it would be interesting to know how the problem on uncovering pairwise nosological relationships compares against a more fine grained prediction of continuous clinical severity indicators (such as the prediction of AQ scores for ASD or the BDI scores for MDD in Section 5.3)

2. Evaluation on Real and Synthetic Data:

(a) With the lack of ground truth optimal dimensional characterization of input examples beyond diagnostic labels, the results of the experiments are a bit hard to interpret beyond consistency over bootstrapped trails. Additionally, it is unclear how the intrinsic dimensionality of the optimal latent space is chosen and how this choice affects performance.

(b) While the consistency of the cluster separation (among various disorders) is examined for their proposed model, the authors restrict the comparison with baselines to a qualitative assessment of the latent embeddings in Appendix C, rather than the full suite of quantitative metrics. They also do not perform these comparisons on ablated versions of their model. Thus, it is still a bit unclear what value the technical sophistication of the model adds to the real world studies.

(c) It is unclear why the authors abruptly chose to report the FID distance metric in the experiments on real data alone. It seems to me that including this metric for the synthetic experiments could help illustrate the fidelity of the recovered clustering in the latent space better.

(d) In Fig. 4, the statistical comparisons on the UTO dataset seem to not reach significance in terms of the proposed pairwise distance relationships they seek to uncover. At the same time, the authors subsequently argue about the consistency in the three way dimensional characterization without adequate explanation for these observations.

3. Validation Procedure:

The authors mention a 10 fold cross validation that is performed  on both datasets (UTO and HuShoWa). They also mention that “Regarding the scarce sample sizes of two datasets, the complex cross validation procedure may not be appropriate here.” Thus, it is unclear how they set various hyperparameters for their model.

4. Clinical Interpretability:

The proposed model seems to offer limited clinical interpretability in terms of identifying the neural correlates or aberrant connectivity patterns that inform the uncovered nosological relationships. The discussion in Section 5.3 is largely heuristic in nature.

(a) The authors restrict this discussion to an associative analysis of the embeddings onto selected disease phenotypes, and identifying a ‘significant connection’ by back projecting from the low dimensional spaces. It is unclear how the significant connection between the Caudate and Cuneus was identified as having the highest impact on discriminating SCZ, ASD and MDD?

(b) Similarly, it is also unclear what the authors mean by isomorphism between the low-dimensional embedding and high-dimensional FC space?

Overall, this discussion suggests that the proposed framework is somewhat limited in scope and clinical applicability.


**Summary Of The Paper:**

The authors propose a novel variational autoencoder to utilize functional connectivity (FC) features from resting state fMRI (rs-fMRI) scans in order to uncover latent nosological relationships between diverse yet related neuropsychiatric disorders. The autoencoder  seeks to learn a mapping  from high-dimensional FC space to a low-dimensional embedding space that is constrained to preserve pairwise relationships between the diagnostic attributes of the disorders.
From a clinical standpoint, this work studies the nosology of complex disorders using a continuous dimensional characterization rather than using categorical and discrete diagnostic labels for supervision.

The authors validate their framework on both synthetic data and two separate clinical rs-fMRI datasets consisting of patients diagnosed with Autism Spectrum Disorder (ASD), Major Depressive Disorder (MDD) and Schizophrenia (SCZ). Their experiments evaluate the consistency of uncovered pairwise nosological relationships inferred from their latent representations.  They demonstrate that their model is capable of reliably inferring a dimensional characterization of multiple brain disorders beyond diagnosis labels.


**Summary Of The Review:**

While the model  and problem being studied is interesting and offers some degree of novelty both in the technical and clinical space,  there are still several concerns with the evaluation and validation being performed. Additionally, the clinical scope of the proposed framework is also not as well justified. For these reasons, I choose to rate the paper as marginally below the acceptance threshold.

---

### Official Review · Reviewer_odDt · 2021-11-03

**Correctness:** 3
**Technical Novelty And Significance:** 3
**Empirical Novelty And Significance:** 3
**Recommendation:** 6
**Confidence:** 2

**Main Review:**

Minor comments:
``Letting N to indicate" should be ``Letting N indicate".

**Summary Of The Paper:**

This paper proposes a novel type of conditional variational auto-encoder that incorporates dual utilization of diagnostic information in learning an optimal embedding space for high dimensional functional connectivity features. The approach is further implemented on two empirical neuropsychiatric neuroimaging datasets to discover a consistent nosological relation among autism spectrum disorder, major depressive disorder, and schizophrenia.

**Summary Of The Review:**

The paper is generally very well written, clearly presented, and a pleasure to read. The technique look sound and the contributions look solid.

---

### Official Review · Reviewer_bgiV · 2021-11-04

**Correctness:** 3
**Technical Novelty And Significance:** 3
**Empirical Novelty And Significance:** 3
**Recommendation:** 6
**Confidence:** 3

**Main Review:**

Strengths:

- The paper presents a new conditional variational autoencoder model.

- The approach uses the diagnostic information in 2 different ways to improve learning the low dimensional embedding.

- Experiments include both synthetic (i.e. with ground truth) and real datasets.


Weaknesses:

- In the experiments, the authors compare to ablation versions of their own model and a few other models. I'm curious whether the authors have tested other conditional VAE models that the authors mention in the introduction (.e.g, CVAE by Sohn et al., 2015)? Also, whether they have tested mix of other models with their proposed "manner 2" loss, e.g. add the contrastive loss to the tested S-VAE.

- The results in Fig 2 look very nice, but I am wondering how stable the results are? That is, across runs does the proposed method consistently produce the correct order of clusterings?

- For the experiments on the real dataset, it is unclear exactly how the cross-validation results are presented. In Fig. 3 left panel, are the displayed clusterings from one fold of cross validation? In the right panel, are the box and whisker plots showing the FID values for the 10 folds?

- While I appreciate the use of 2 datasets, I am wondering whether model generalization was tested, i.e., using the trained model from UTO, did the authors try testing that model on the HuShoWa dataset, and vice versa? Such generalization is really important in neuroimaging domain, as gathering of large datasets often require bringing together data across imaging sites.

- Suppose the authors can find a consistent nosological relationship across neuropsychiatric disorders. Could the authors comment on the significance of achieving such an ordering, in terms of clinical relevance?

- The writing in the paper needs improvement - there are many long and awkward sentences/phrasings that make it difficult sometimes to understand the authors' points.

Minor comments:

- Could the authors clarify whether the compared NCA uses any signal (ie diagnosis) supervision or is purely unsupervised? If purely unsupervised the comparison seems a bit unfair.

- In Fig. 3 middle lower panel, the FID between MDD-SCZ is very small, compared to what we see visually for those clusters compared to other paired clusters - wanted to check if this reporting is correct?



**Summary Of The Paper:**

This paper presents a new conditional variational autoencoder (VAE) approach to learn a low dimensional embedding of neuropsychiatric disorders from resting-state functional connectivity data. The proposed approach uses diagnostic information in 2 ways: 1) to cluster samples from the same disorder together via the conditional VAE model, and 2) to separate the clusters using contrastive learning.  The method is compared against other dimension reduction approaches on both a synthetic dataset and 2 real datasets which showed a consistent nosological relation among 3 disorders.

**Summary Of The Review:**

My recommendation is based on the proposed new type of conditional VAE, which could be applied to any other type of dataset, and the fairly convincing results on both synthetic and real datasets. However, I am unsure exactly of the clinical relevance of the paper's stated purpose (of finding nosological relationships between disorders), and I feel a few relevant comparisons are missing.

---

### Decision · Program_Chairs · 2022-01-20

**Decision:**

Reject

**Comment:**

This paper proposes a novel variational autoencoder to utilize functional connectivity (FC) features from resting state fMRI (rs-fMRI) scans in order to uncover latent nosological relationships between diverse yet related neuropsychiatric disorders. The methodology and main technical contributions are clearly articulated and explained, and the experimental results seem convincing. On the other hand, the proposed framework is somewhat limited in scope and clinical applicability, and the writing in the paper needs improvement (as pointed out by two reviewers).